# Morphological characterization and DNA barcoding of *Ruellia* sp. in Saudi Arabia

**Jawaher Saad Almuteri[1,2], Mona Soliman Al wahaibi[1], Abd El-Zaher Mohamad Mustafa[1], Manal Abdullah Alshaqhaa[3], Muhammed Afzal[4], Manar Dhafer Alshehri[3]***

**1** Botany and Microbiology Department, College of Science, King Saud University, Riyadh, Saudi Arabia, **2** Applied Sciences Department, College of Science, Northern Border University, Arar, Saudi Arabia, **3** Biology Department, College of Science, King Khalid University, Abha, Saudi Arabia, **4** Department of Plant Production, College of Food and Agricultural Sciences, King Saud University, Riyad, Saudi Arabia

* mnar@kku.edu.sa

**Data Availability Statement:** All relevant data are within the paper and its Supporting Information files.

## Abstract

The genus *Ruellia* L. belongs to this family and its plants are herbs or shrubs. This genus was first detected in the tropical and subtropical regions. The primary objective this study is to identify the various species within the genus *Ruellia* using both morphological characteristics and DNA barcoding methods. For this purpose, plant samples were meticulously collected from eight distinct natural habitats across the region. All vegetative and floral parts were examined using a binocular microscope. All parts are measured and photographed. To ensure accurate identification and characterization, four molecular barcoding markers were employed: *Psbk-psbi*, *trnH-psbA*, *rbcL, and AtpF-AtpH*. Eight *Ruellia* species were identified from different regions: Abha, Aseer (24651), Jazan (24652), Malacosperma, Patula, Taif (24650 rose), Taif (24650 violet), and Taif (24650 white). The species were confirmed using specimens from the King Saud University herbarium. Notably, the samples collected from Taif, which had flowers of different colors, were determined to represent a single species with different genotypes. The use of four DNA barcode markers (*Psbk-psbi*, *trnH-psbA*, *rbcL, and AtpF-AtpH*) facilitated the identification of five distinct species: *R. tweediana*, *R. sp. SH2010*, *R. carolinensis*, *R. simplex*, and *R. patula*. These findings confirmed the dominant *Ruellia* species in Saudi Arabia and demonstrated the reliability of DNA barcode markers for species identification. Further assessment of these species' adaptability, molecular genetics, and functional genomics is necessary for their commercial utilization in the region. These species are recorded for the first time in Saudi Arabia and represent the first record.

## Introduction

The family Acanthaceae is predominantly found in Indonesia, Malaysia, Africa, and Central America, with 24 genera and 35 species represented in the Kingdom of Saudi Arabia [1]. Among these, the genus *Ruellia* L. includes a variety of herbs and shrubs, primarily distributed in tropical and subtropical regions, encompassing approximately 350 species worldwide [2]. *Ruellia* species are of notable medicinal importance, particularly *R. tuberosa*, which contains

**Funding:** Deanship of Research and Graduate Studies at King Khalid University submitted the funding for this work through small group research under grant number RGP1/200/45.

**Competing interests:** The authors have declared that no competing interests exist.

benzoxazinoids with applications in treating stomach cancer and leukaemia [3–5]. These compounds are also utilized for skin diseases, ocular ailments, and bronchitis [6–8]. *Ruellia* species are significant in traditional medicine due to their diverse phytochemical compounds, including flavonoids, alkaloids, and coumarins, which exhibit various pharmacological activities such as wound healing, anti-diabetic effects, and anti-inflammatory properties [5, 9, 10]. Despite their medicinal importance, comprehensive taxonomic studies on *Ruellia* in Saudi Arabia are lacking. Previous reports have identified only a few species, and many remain unclassified, leading to an incomplete understanding of this genus within the region [11, 12]. It also contains compounds that are important as insecticides and anthelmintics [13]. Additionally, the herbarium at King Saud University houses numerous unidentified specimens. This highlights a substantial research gap, as accurate identification is crucial for conservation and medicinal use.

To address this gap, our study employs DNA barcoding, a technique increasingly recognized for its ability to facilitate precise species identification [14]. The genetic barcoding of DNA is very important for researchers in the field of taxonomy [15]. DNA barcoding will help to identify known species, discover thousands of species yet to be named, and manage the vast biodiversity on Earth [16]. We focus on specific plastid DNA barcodes, notably the *rbcL* and *matK* genes, as well as the *trnH-psbA* intergenic spacer. These markers were chosen based on their proven efficacy in plant taxonomy [17, 18], providing a robust framework for distinguishing closely related species. While other barcoding primers, such as those from chloroplast-plastid and nuclear regions, are valuable, our focus on these plastid genes aligns with current literature demonstrating their effectiveness in resolving phylogenetic relationships within the *Acanthaceae* [18], including the genus *Ruellia* based on large plastid genes. They found that *rbcL* is more efficient, while a combination of *matK* and *trnH-psbA* can be used to identify and study the phylogenetics of plants. No previous taxonomic studies on the plants of the genus *Ruellia* in KSA were found. This study employed DNA barcoding, in addition to phenotypic description, on several plant samples from the genus *Ruellia*. DNA barcoding was chosen for its capacity for accurate plant identification [14] owning to its high quality [19].

This study utilizes DNA barcoding technology to identify previously unknown species of the genus *Ruellia* through specific plastid DNA sequences, which provide reliable and high-quality plant identification [14, 19]. This method enhances the ability to distinguish closely related species and has evolved with advancements in gene sequencing, allowing for the use of complete chloroplast genomes [14]. DNA barcoding operates similarly to traditional taxonomy but offers greater speed and resolution in species identification, using standardized genomic regions that are universally present across targeted lineages. By generating DNA sequences from small tissue samples and comparing them to reference libraries like BOLD, researchers can achieve rapid and reproducible identifications [20]. This technique is crucial for taxonomists as it accelerates the discovery of new species and improves our understanding of biodiversity [15, 16] Common barcodes in plants include the *rbcL* and *matK* genes, which have proven effective for phylogenetic studies within the Acanthaceae family [21]. Developing effective DNA barcodes for plants is challenging. However, both single-locus barcodes (like *rbcL*, *ITS*, *ITS2*, *matK*, *rpoB*, *rpoC*, and *trnH-psbA*) and multi-locus barcodes have proven successful for species-level identification of herbal plants. Multi-locus barcoding, in particular, has also become an efficient method for authenticating herbal products [22]. The implications of DNA barcoding extend beyond species identification; it serves as a pivotal tool for biodiversity management, enabling rapid assessments and discovery of new species [16]. Enhanced accessibility of complete chloroplast genomes has further improved the resolution of this methodology, promising a transformative impact on our understanding of plant diversity [14]. Following initial assessments of various plastid markers for plant barcoding, including both coding and non-coding regions [23, 24], three research groups proposed four main barcoding

strategies. These involved different combinations of seven plastid markers: *rpoC1+rpoB +matK*, *rpoC1+matK+trnH-psbA* [25], *rbcL+trnH-psbA* [26], and *atpF-H+psbK I+matK* [27]. The current study also employed a comprehensive analysis of various combinations of barcode markers, integrating multiple approaches to assess their effectiveness and optimize the accuracy and reliability of plant species identification. Through our research, we aim to contribute to the taxonomic resolution of *Ruellia* species in Saudi Arabia, ultimately aiding in conservation efforts and promoting the utilization of these plants in traditional medicine. By integrating morphological and molecular approaches, this study seeks to provide a comprehensive overview of the genus, bridging significant gaps in existing literature.

## Materials and methods

### Plant samples

Samples were collected from their regions in Saudi Arabia, (Madinah, Abha, Aseer, Riyadh, Jazan, Taif). All vegetative and floral parts were examined using a binocular microscope. All parts are measured and photographed. then dried and a copy of them was preserved in the herbarium of King Saud University. One sample of leaves was taken from each plant separately and used to identify the plant using barcode technology. Detailed information about the collection locations and herbarium numbers of each plant sample presented in Table 1 and analyse the of morphological results numerically using Microsoft XLSTAT.

The plant samples used in this study were collected from eight natural habitats (Fig 1). We examined the shoot system, including the stem and leaves, focusing on the blade shape, margin, base, and tip. Additionally, we analysed the inflorescence, flower density along the axis, calyx, corolla, stamens, and pistils.

### DNA extraction

DNA was extracted from specimens by crushing the tissue in liquid nitrogen using the CTAB method [28]. The acquired pellet was cleaned with 70% ethanol and permitted to dry at room temperature. DNA pellet suspended in Tris Ethylenediaminetetraacetic acid (TE) buffer was quantified by using a UV Spectrophotometer with A260/280 ranges from 1.62–1.78. The detail information of each primer used in the study mentioned in Table 2.

### PCR amplification

A PCR reaction was carried out in 0.2 ml PCR tubes by using 25μl total volume to amplify the desired markers. The valuable loci were amplified by using a robust PCR kit (Applied Bio

**Table 1. List of plant samples of the genus *Ruellia* studied and collection sites in Saudi Arabia.**

| Taxa | Collection place | Collection date | GPS coordinates |
|---|---|---|---|
| 1. *Ruellia sp.Abha* | Abha | December, 2021 | 18˚16'37.6"N 42˚43'23.5"E |
| 2. *Ruellia sp.24651*[*] | Aseer | December, 2021 | 19˚07'44.5"N 41˚55'42.2"E |
| 3. *Ruellia sp.24652*[*] | Jazan | September, 2021 | 16˚59'30.7"N 42˚42'59.0"E |
| 4. *Ruellia malacosperma* | Riyadh, Jazan, Madinah | September, 2021 | 24˚43'07.9"N 46˚37'24.4"E |
| 5. *Ruellia patula* | Jazan | September, 2021 | 16˚54'50.1"N 42˚33'20.4"E |
| 6. *Ruellia.sp.24650rose*[*] | Taif | December, 2021 | 21˚30'34.8"N 40˚29'17.2"E |
| 7. *Ruellia.sp.24650.* Violet[*] | Taif | December, 2021 | 21˚30'34.8"N 40˚29'17.2"E |
| 8. *Ruellia.*sp.24650 White | Taif | December, 2021 | 21˚30'34.8"N 40˚29'17.2"E |

[*] Herbarium number

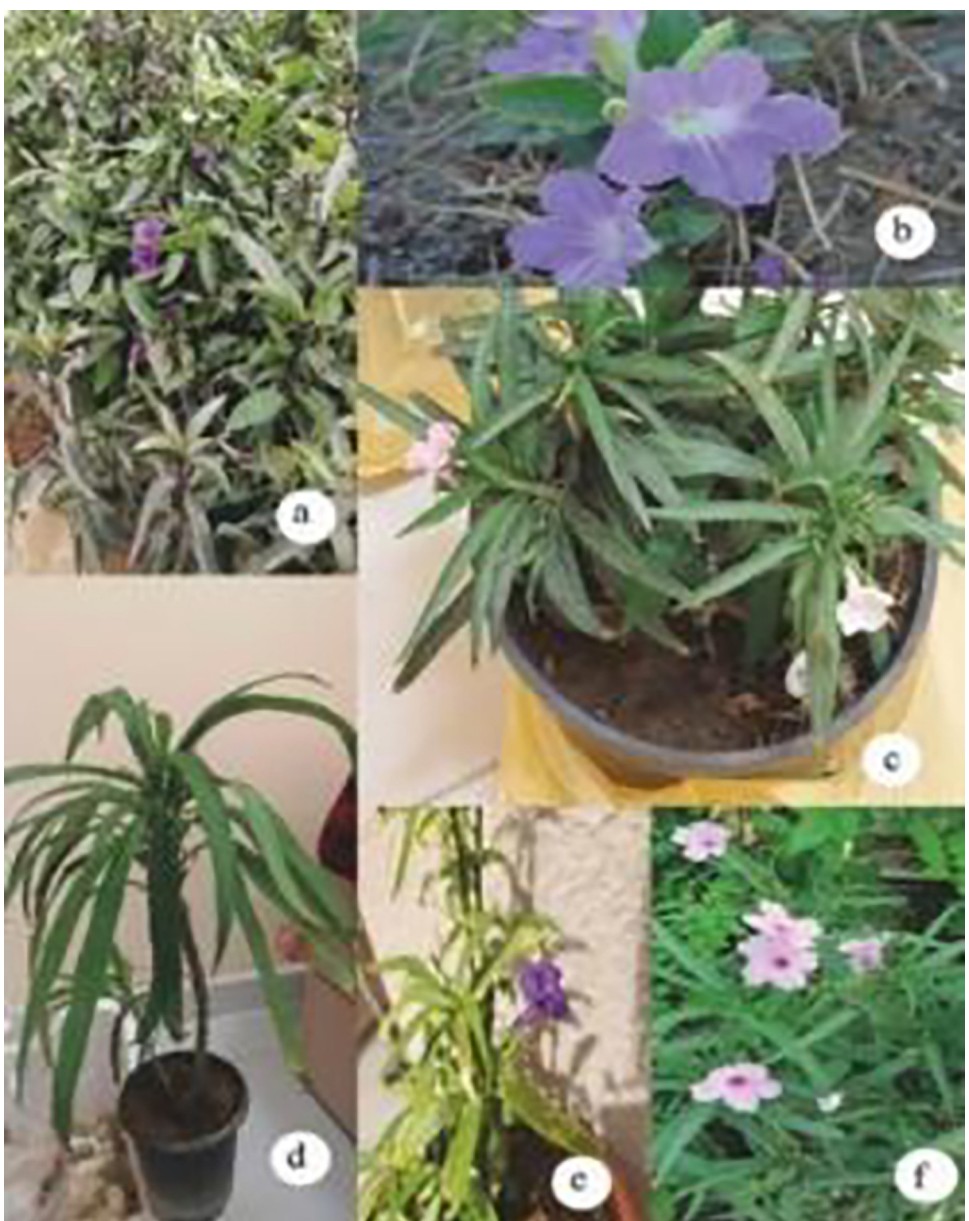

**Fig 1.** Plants in their apparent shape (a) *R. malacosperm* (b) *R. patula* and *R. sp.24651*. (c) *R. sp.24650 rose* and *R. sp.24650* White (d) *R. sp Abha* (e) *R.sp.24650*. Violet (f) *R. sp.24652*.

**Table 2. Primer name, sequences, and Tm value.**

| Primer name | Forward Seq. | Reverse Seq. | Tm value ($^{O}$C) | Reference |
|---|---|---|---|---|
| psbK-psbI | TTAGCATTTGTTTGGCAAG | AAAGTTTGAGAGTAAGCAT | 55–60 | 42 |
| trnH-psbA | GTTATGCACGAACGTAATGCTC | CGCGCGTGGTGGATTCACAATCC | 55–60 | 42 |
| atpF-atpH | ACTCGCACACACTCCCTTTCC | GCTTTTATGGAAGCTTTAACAAT | 60–65 | 42 |
| rbcL | GTAAAATCAAGTCCACCACG | ATGTCACCACAAACAGAGACTAAAGC | 60–65 | 51 |

systems, Foster, California, USA). For the *matK*, the cycling conditions were 95˚C for 10 minutes, 30 cycles at 95˚C for the 30s, 55˚C for 45s, 72˚C for 1min: and the final elongation at 72˚C for 4min. The rbcL primers were cycled at 95˚C for 10 min, 30 cycles at 58˚C for the 30s, 72˚C for 1 min, and finally for 4 min, at 72˚C for final elongation. PCR products were separated were visualized on 1% agarose gel and purified through a DNA purification kit (Thermo Scientific, Waltham, Massachusetts, USA).

## Sequencing

The purified PCR samples were sent to Macrogen in South Korea (https://www.macrogen. com/en/main/index.php, accessed on 15, June 2022) for sequencing using an ABI3730 automated sequencer with the same markers as in the PCR reactions. Both DNA strands were sequenced and checked for ambiguous nucleotides.

## Data analysis

The first 30 base pairs of each read were removed, the length of the amplicon products was measured, and multiple DNA sequence alignments were generated using ClustalW alignment software [29]. The phylogenetic tree was constructed using MEGA X to group the clones with following matrix: DNA weight matrix = IUB, transition weight = 0.50, delay divergent cutoff 30% [30]. The evolutionary history was inferred by using the maximum likelihood method and the Tamura–Nei model [30]. The evolutionary distances were computed using the maximum composite likelihood method and are in the units of the number of base substitutions per site. Evolutionary analyses were conducted in MEGA X using the Matrix Representation [20] with Parsimony method [31]. The numbers on the nodes represent the bootstrap values as percentages from 1000 replicates.

## Results

### Morphological characteristics

The morphological characteristics of various *Ruellia* species identified in the study were discussed (S1 Table). Each row corresponds to a different species or subspecies, detailing their specific traits such as plant height, color, stem characteristics, leaf shape, pedicel height, calyx shape, corolla color, and fruit features (Fig 2).

### DNA barcoding

The Table 3 offers a detailed overview of the genetic discrimination of various *Ruellia* species collected from different locations (Madinah, Abha, Aseer, Riyadh, Jazan and Taif) in KSA. The analysis aims to accurately identify the species by matching specific genetic markers (S2 Table). Four DNA barcode markers were used: *Psbk-psbi*, *Trnh-psbA*, *Atpf-Atph*, and *RbcL*. For each marker, the table lists the matched species and the corresponding accession number (Table 3). The collection from Abha identified by two primers, *Psbk-psbi* and *AtpF-AtpH*, identified the species as *R. brittoniana*, while *TrnH-psbA* identified it as *R. simplex* and *Rbcl* identified it as *R. tweediana*. Similarly, the sample *Ruellia* sp. 24651 was identified as *R. brittoniana* by *Psbk-psbi and AtpF-AtpH*, as *R. patula* by *TrnH-psbA*, and as *R. sp. sh2010* by *rbcl*. The sample *Ruellia sp. 24652* was consistently identified as *R. brittoniana* by three markers and as *R. carolinensis* by one marker. Similarly, *Ruellia malacosperma* was identified as *R. brittoniana* by three markers and as *R. sp. sh2010* by one marker. *Ruellia patula* was identified as *R. brittoniana* by three markers and as *R. patula* by one marker. *Ruellia sp. 24650 rose* was consistently identified as *R. brittoniana* by three markers and as *R. tweediana* by one marker. *Ruellia*

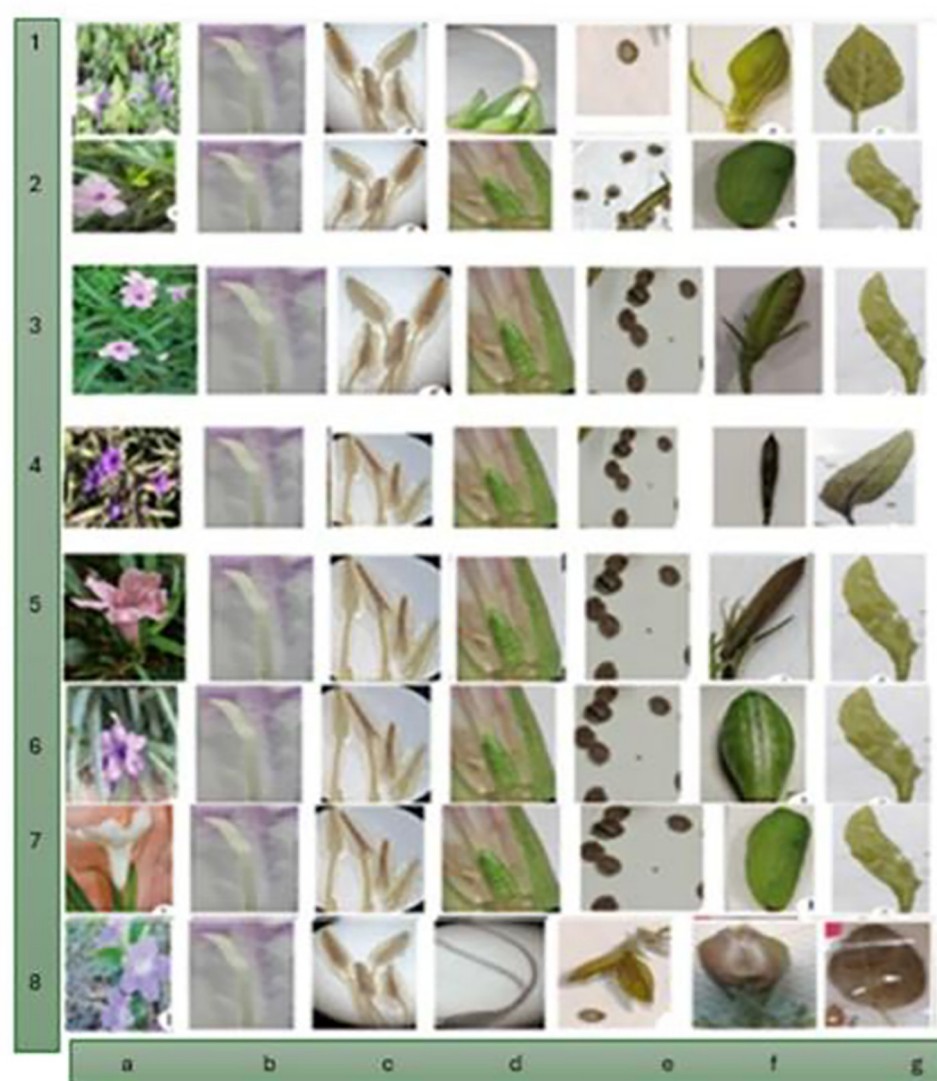

**Fig 2.** (a) flower shape, corolla (b) stigma, style (c) stamens (d) ovary (e) seeds (f) fruits (g) leaf. 1- *R.sp.24651- R.sp. sh2010* 2- *R.sp.Abha-R.tweediana* 3- *R.sp.24652- R.carolinensis* 4- *R.simplex -R.malacosperma* 5- *R.sp.24650-rose-R. tweediana-rose* 6- *R.sp.24650-violat-R.tweediana-violat* 7- *R.sp.24650-white-R.tweediana-white* 8- *R. patula.*

sp. 24650 violet was identified as *R. brittoniana* by three markers and as *R. tweediana* by one marker. *Ruellia* sp. 24650 with was identified as *R. brittoniana* by one marker and as *R. tweediana* by two markers. The application of barcode markers estimates the genetic diversity and highlights potential identification issues within *Ruellia* species, demonstrating the importance of using multiple barcoding markers for accurate species identification.

The genetic analysis of *Ruellia* species collected from Abha and various other sites revealed diverse identifications based on different genetic markers. For the Abha collection, two primers (*Psbk-psbi* and *AtpF-AtpH*) identified the species as *R. brittoniana*, while *TrnH-psbA* identified it as *R. simplex* and *rbcl* as *R. tweediana*. Similarly, *Ruellia sp.* 24651 was identified as *R. brittoniana* by *Psbk-psbi* and *AtpF-AtpH*, as *R. patula* by *TrnH-psbA*, and as *R. sp. sh2010* by *rbcl*. Other samples, such as *Ruellia sp.* 24652, *Ruellia malacosperma*, *Ruellia patula*, and various *Ruellia sp.* 24650 variants, showed consistent identifications by some markers and differing

**Table 3. DNA barcoding results and their accession numbers.**

| Collection Site | *Psbk-psbi* | *Trnh-psba* | *Atpf-Atph* | *RbcL* |
|---|---|---|---|---|
| *Ruellia sp.Abha* | *R. brittoniana* (OP893975) | *R. simplex* (OP681139) | *R. brittoniana* (OP893974) | *R. tweediana* OP480015 |
| *Ruellia sp.24651* | *R.brittoniana* (OP503876) | *R.patula* (OQ095390) | *R.brittoniana* (OP503877) | *R.sp.sh2010* (OP867066) |
| *Ruellia sp.24652* | *R.brittoniana* (OP745454) | *R.simplex* (OP947575) | *R.brittoniana* (OP745452) | *R.carolinensis* (OP867067) |
| *Ruellia malacosperma* | *R.brittoniana* (OP745458) | *R.simplex* (OP745457) | *R.brittoniana* (OP745456) | *R.sp.sh2010* (OP745455) |
| *Ruellia patula* | *R.brittoniana* (OP480013) | *R.patula* (OQ106916) | *R.brittoniana* (OP480014) | *R.brittoniana* (OP745453) |
| *Ruellia sp.24650.rose* | *R.brittoniana* (OP745448) | *R.simplex* (OP745449) | *R.brittoniana* (OP745450) | *R.tweediana* (OP745451) |
| *Ruellia sp.24650violat* | *R.brittoniana* (OP712209) | *R.simplex* (OP712210) | *R.brittoniana* (OP712211) | *R.tweediana* (OP712208) |
| *Ruellia sp.24650.withe* | *R.brittoniana* (—) | *R.tweediana* (OP712207) | *R.brittoniana* (OP745447) | *R.tweediana* (OP712206) |

identifications by others. This highlights the genetic diversity and potential identification challenges within *Ruellia* species, underscoring the importance of using multiple barcoding markers for accurate species identification (Fig 3).

The two markers (*psbK-psbI* & *AtpF-AtpH*) showed similar identification of the *Ruellia* sp, however, the other two primers *TrnH-psbA* & rbcl) showed diverse results. For that purpose, wo combine the sequence of *TrnH-psbA* & rbcl and construct the dendrogram to determine the similarity of the isolates (Table 4; Fig 4). The combined sequence analysis identified several *Ruellia* species, including *Ruellia* sp. *Abha* (*R. tweediana*), *Ruellia* sp. 24651 (R. sp. sh2010), *Ruellia* sp. 24652 (*R. carolinensis*), *Ruellia malacosperma* (*R. simplex*), *Ruellia patula* (*Ruellia patula*), *Ruellia* sp. *24650.rose* (*R. tweediana* rose), *R. tweediana* violet (*Ruellia sp. 24650 violet*), and *R. tweediana* white (*Ruellia sp. 24650 white*). The combination of two different marker regions provided more accurate information about the species, allowing for precise identification and classification (Fig 5).

## Discussion

This study revealed that the herbaceous, perennial, and branching *Ruellia* species collected from the KSA are consistent with previous studies. The findings align with [32], in type species [12] and Chaudhary's report on *Ruellia patula* [33]. Similarly, on [12, 34], reported similar observations on *Ruellia sororia*, which Kew Garden refers to as synonym with *Ruellia brittoniana* were corroborated. The study also indicates that stem and leaf hairiness among species, with *R. patula* and *R. sp. 24651* having stems completely covered in hair.

This study showed that the plants of *Ruellia* species collected from the KSA are herbaceous, perennial, and branching, consistent with the previous research. These findings align with [11, 12, 32], on *Ruellia patula*, as well as [12, 33, 34], on *Ruellia sororia*, which Kew Garden refers to as the synonym with *Ruellia brittoniana*. The study also found that stem and leaf hairiness vary among different species, with stems of *R. patula* and *R. sp.24651* being completely covered in hairs. This result agreed with [35] in his study of *Ruellia asperula*. Additionally, some species, such as *R. sp. Abha*, and *R. sp.24650* (rose-white) have prominent nodes. This result aligns with, [19] study of the type *Ruellia tweediana*. The current study also showed that the leaves of all the examined species are simple, petiole and complete [12]. However, leaf shapes and edges vary among species. They range from lanceolate in *R. malacosperma* and *R. sp. Abha* [36], to

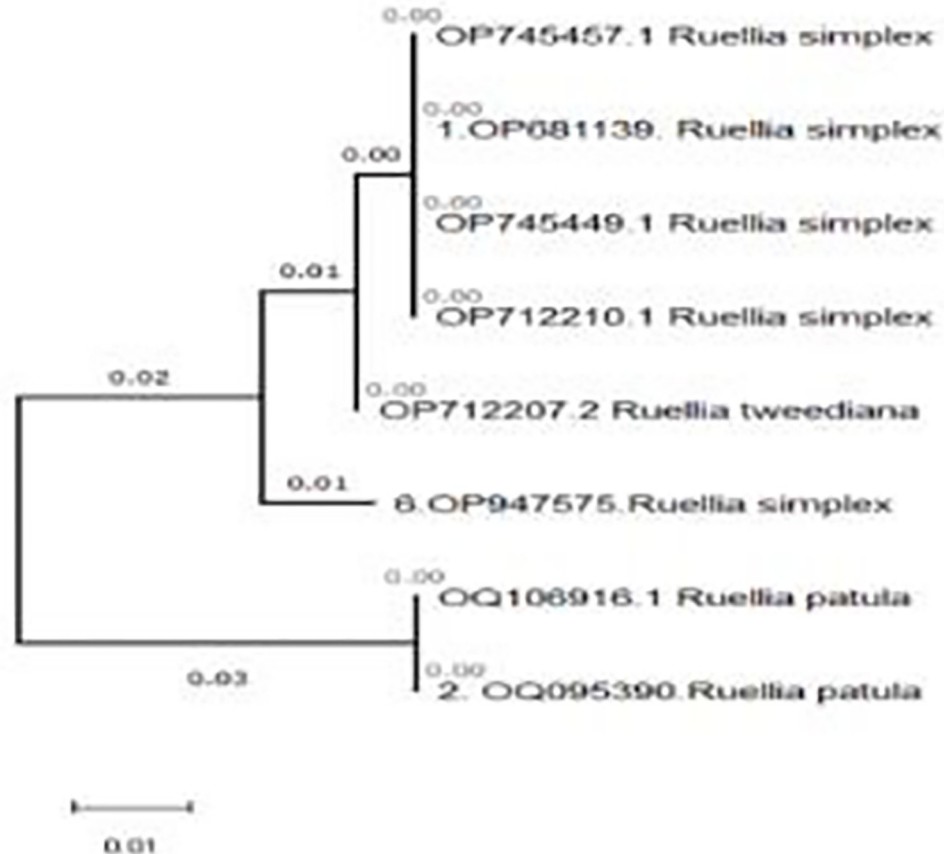

**Fig 3. Phylogenetic tree constructed based on concatenated *Ruellia sp. sequences* along with reference sequence from NCBI using maximum likelihood and neighbor-joining with 1000 bootstrap value with Maga X.**

heart-to-ovate in *Ruellia patula* [37]. also observed oval shapes in *R. sp.24651* and bar shape in *R. sp.24652* and *R. sp.24652.R. sp.24650* (rose-violet-white) consistent with this results. The presence, quantity, and distribution of hairs on the leaves, with some species like *R. patula* and *R. sp. 24651* having entire leaves covered in hairs, while others like *R. malacosperma* have hairs concentrated on midrib and leaf margins. This observation agrees with [12], findings in his study of *Ruellia patula*, *Ruellia malacosperma*, *Ruellia prostrata* and *Ruellia grandiflora*.

**Table 4. identification results after combining both molecular markers (*Trnh-psba* and *rbcl*).**

| Taxa | Taxa after combining (*Trnh-psba* and *rbc*l) |
| --- | --- |
| *Ruellia sp.Abha* | *R.tweediana* |
| *Ruellia sp.24651* | *R.sp.sh2010* |
| *Ruellia sp.24652* | *R.carolinensis* |
| *Ruellia malacosperma* | *R.simplex* |
| *Ruellia patula* | *R.patula* |
| *Ruellia sp.24650.rose* | *R.tweediana.rose* |
| *Ruellia sp.24650.violat* | *R.tweediana.violat* |
| *Ruellia sp.24650.withe* | *R.tweediana.white* |

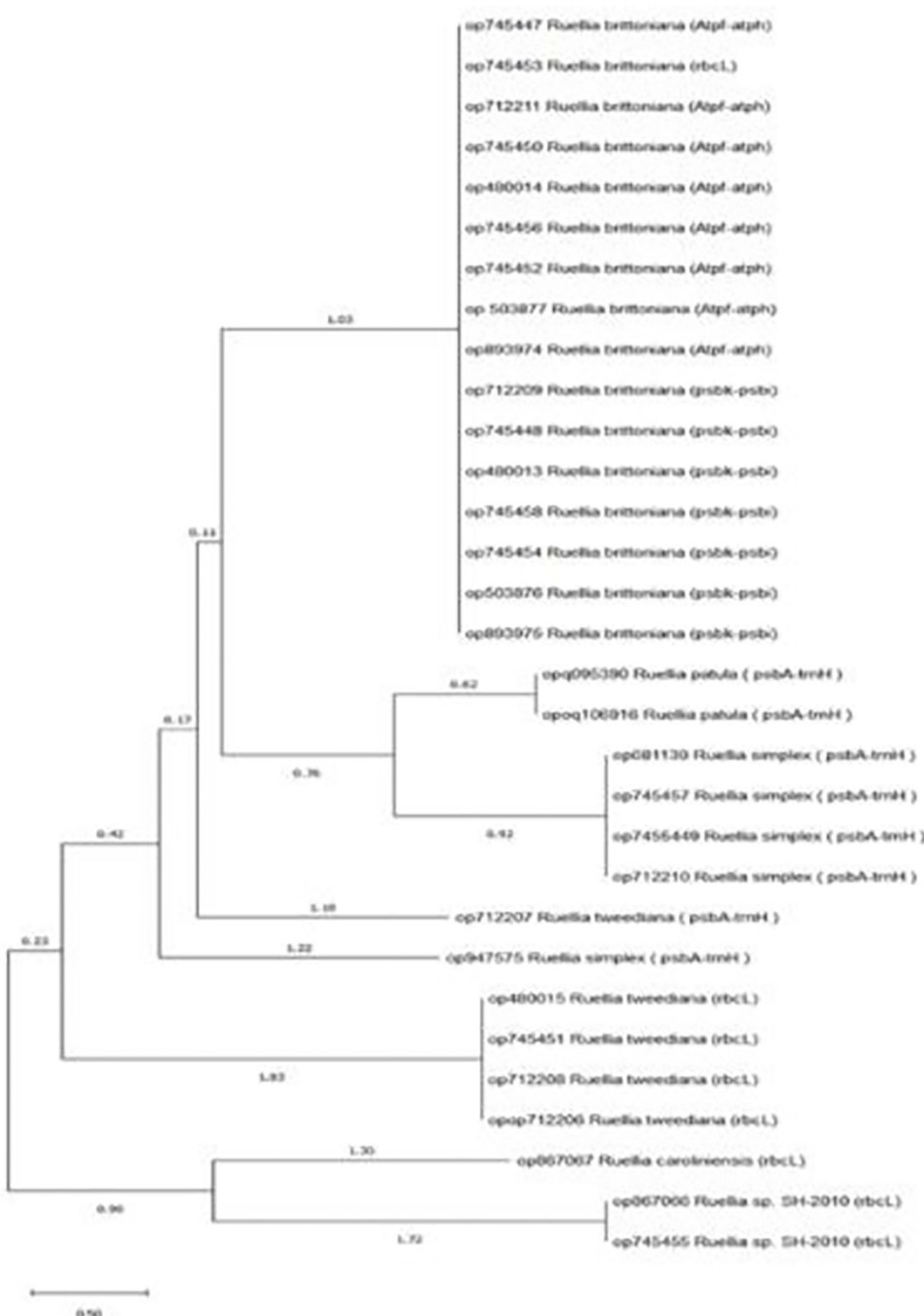

**Fig 4. Combine phylogenetic tree constructed using Maga X with 1000 bootstrap value of concatenated *Psbk-psbi*, *Trnh-psba*, *Atpf-Atph* and *rbcL* generated sequences from *Ruellia* sp and their represented sequences from NCBI.** The percentage of bootstrap value from 1000 replicates is represented by the number of nodes. The horizontal bar represents the genetic distance.

The plants collected in this study showed that the flowers in all types of *Raullia* are arranged on the stem in limited, two-branched inflorescences. They are five-parted, single, symmetrical, hermaphroditic, funnel-shaped, covered with glandular hairs. Color varied among species: pale pink in *R. malacosperma*, pale pink color in *R. sp.24652*, dark pink color in *R. sp.24650. rose*, white color in *R. sp.24650.*white and a mixture of white with violet color in *R. patula and*

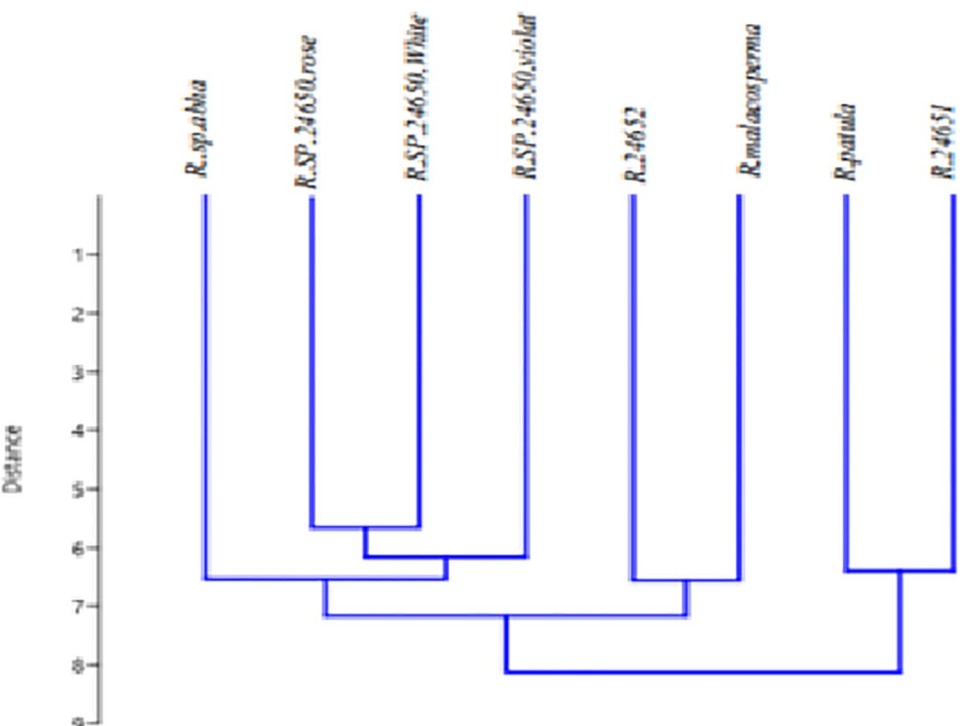

**Fig 5. Dendrogram of the species under study based on the characters Eucledean Morphology.**

*R. sp.24651*. This finding corroborates [38], *Ruellia brittoniana* [33], *Ruellia tweediana* [39] and study on *Ruellia simplex [40]*. Additionally, stems and leaves of some species, such as *R. malacosperma* and *R. sp. 24650* (violet), turn purple when exposed to high temperatures, as observed by [33]. All analyzed samples exhibited a consistent cup with fruit and distinct hairy sepals, featuring transparent membranous margins. Shapes varied, ranging from lanceolate in *R. sp. Abha* to threadlike *in R. patula and R. malacosperma—R. sp. 24652*, and ovoid in species *R. sp. 24650* (white, rose, violet) and *R. sp. 24651*. This finding aligns with study on *R. brachysiphon* and *R. angustiflora* species [41], as well as with [12] and research on *Ruellia jiboia* [42].

The flowers have five fused petals, funnel-shaped, with different colors, which differ according to the type, and the reason may be due to the different colors of the corolla in response to the amount of sunlight and fertilizers they get in the soil. The result of this study is consistent with the study of *R. simplex* [39].

All the studied species contained four smooth stamens above a petal, fertile, of the long-two type. As for the pistil in all species, it consisted of an ovary comprising two lodgings, and the ovules were in an axial placental position, elliptical in all species except for *R.sp.24651*, where the ovary appeared. It is spherical to ovoid in shape, at the base of the ovary there is a nectar disc, and the stigma is single, covered with hairs, and the stigma is in two parts and takes on a white to pale pink color.

The fruit is a simple dry open capsule, ranging from ovoid to spherical in *R. patula-R. sp. 24651*, and oblong-ovate in *R. sp. 24650* (rose-violet-white), *R. sp. 24652*, and *R. sp. Abha*, with an elongated oval shape in *R. malacosperma*. All fruits exhibit an upper prominence, initially green before ripening, and turning brown upon ripening. The seeds inside are equipped with hooks, which, upon exposure to moisture, cause the fruit to burst, dispersing the seeds [34, 41, 43].

Proposed molecular barcoding markers such as *matK*, *rbcL*, and *psbA-trnH* that exhibit a higher rate of nucleotide substitutions, making them one of the most rapidly evolving protein-coding regions of chloroplast DNA (cpDNA) [21]. However, molecular markers are evolving more rapidly than coding regions, with *psbA-trnH* being one of the fastest evolving markers in chloroplast DNA (cpDNA) [8], and the most viable candidate for obtaining a single-site barcode for plants [44]. The perfect standard barcode sequence is still not found in plants so a combination of two or more plastid barcodes used to identify and distinguish species [21], and it is worth noting that both *R.tweediana* and *R.simplex* species, *R.brittoniana* and *R.malacosperma* are only one species, according to [24, 33].

The final result that we reached, after coding the DNA of all the studied plant samples, is that the eight species are only five species: *R. tweediana*, *R.sp.sh2010*, *R.carolinensis*, *R.simplex*, *R.patula*, and that the species collected from the region Taif, and it has flowers of different colors. It is only one species, and the difference in the color of the corolla may be due to it in response to environmental changes such as the photoperiod to which the plant was exposed or the soil pH, as mentioned by [39].

Previous genetics studies have reported similar results [2], such as [1, 2, 18, 24, 45, 46]. The former, while studying some species of the Acanthaceae family, suggested that the *rbcl* barcode sequence is more effective, while a combination of *matk*, *trnh-psba*, can be used to study and determine the phylogeny of plants. The *psbA-trnH* trees proved more effective in resolving close species relationships within the genus *Ruellia* compared to other markers. This enhanced resolution might stem from these markers originating from the chloroplast genome and sharing a similar evolutionary history [42].

The latter recommended a combination of the *psbA-trnH* and *rbcl* cassettes as a universal barcode for interspecies discrimination, the *trnH-psbA* cassette will result in valid species counting where *rbcL* lacks discriminatory power, especially in species-rich genera of angiosperms. Both sites have standard primers currently available that make them universally amplifiable with minimal effort in the widest range of wild plants. This two-site plant barcode is now being applied to build a library of more than 700 species of the world's most important medicinal plants [47]. The *trnH-psbA* and *rbcl* sequences were combined to accurately identify the samples. Combining the non-coding *trnH-psbA* sequence with one of three coding regions, *rbcL*, *rpoB2 or rpoC1*, offers the highest the highest comprehensiveness and greater ability to discriminate between species pairs. Complementing a rapidly evolving locus such as the *trnH-psbA* with a more conservative locus such as rbcL can reduce type 1 errors [26]. In this study, we followed this approach by combining both *trnH-psbA* and *rbcL* sequences.

## Conclusion

The species *Ruellia* sp. SH2010 was identified in the Asir region through DNA barcoding using the *rbcL* sequence; however, its exact classification remains uncertain and requires further investigation. This aligns with the study's objective to assess the efficacy of DNA barcoding in the identification of *Ruellia* species in Saudi Arabia. As highlighted by previous research, the *rbcL* marker alone may not provide sufficient resolution for accurate species identification due to the lack of an ideal universal barcode. Therefore, this study emphasizes the importance of using a combination of molecular markers, such as *trnH-psbA*, alongside *rbcL*, to improve the precision of species identification. By integrating multiple markers, the study advances the objective of enhancing the accuracy of both morphological and genetic characterization of *Ruellia* species in the region, contributing valuable insights for future taxonomic investigations.

## Supporting information

**S1 Table. Morphological characteristics of plant species recorded during the study.**
(DOCX)

**S2 Table. Numercial analysis of barcode data.**
(XLSX)

## Acknowledgments

[funding sources should not be included here or in the manuscript file, only during manuscript submission]

## Author Contributions

**Conceptualization:** Jawaher Saad Almuteri, Mona Soliman Al wahaibi, Manal Abdullah Alshaqhaa, Manar Dhafer Alshehri.

**Data curation:** Jawaher Saad Almuteri, Muhammed Afzal.

**Formal analysis:** Jawaher Saad Almuteri.

**Funding acquisition:** Manal Abdullah Alshaqhaa, Manar Dhafer Alshehri.

**Investigation:** Jawaher Saad Almuteri.

**Methodology:** Jawaher Saad Almuteri.

**Project administration:** Jawaher Saad Almuteri.

**Resources:** Jawaher Saad Almuteri.

**Software:** Jawaher Saad Almuteri, Mona Soliman Al wahaibi, Muhammed Afzal.

**Supervision:** Mona Soliman Al wahaibi, Abd El-Zaher Mohamad Mustafa.

**Visualization:** Jawaher Saad Almuteri, Mona Soliman Al wahaibi, Muhammed Afzal.

**Writing – original draft:** Jawaher Saad Almuteri.

**Writing – review & editing:** Jawaher Saad Almuteri, Mona Soliman Al wahaibi, Abd El-Zaher Mohamad Mustafa, Manal Abdullah Alshaqhaa, Muhammed Afzal, Manar Dhafer Alshehri.

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
