## [Decision Letter · Decision Letter 0]

16 Sep 2024

PONE-D-24-30210Morphological characterization and (DNA) Barcoding identification of Ruellia spp. (Acanthaceae) in Saudi ArabiaPLOS ONE

Dear Dr. ALSHEHRI,

Thank you for submitting your manuscript to PLOS ONE. After careful consideration, we feel that it has merit but does not fully meet PLOS ONE’s publication criteria as it currently stands. Therefore, we invite you to submit a revised version of the manuscript that addresses the points raised during the review process.

We look forward to receiving your revised manuscript.

Kind regards,

Abul Khayer Mohammad Golam Sarwar

Academic Editor

PLOS ONE

Journal Requirements:

1. When submitting your revision, we need you to address these additional requirements.Please ensure that your manuscript meets PLOS ONE's style requirements, including those for file naming. The PLOS ONE style templates can be found at https://journals.plos.org/plosone/s/file?id=wjVg/PLOSOne_formatting_sample_main_body.pdf and https://journals.plos.org/plosone/s/file?id=ba62/PLOSOne_formatting_sample_title_authors_affiliations.pdf 2. In your Methods section, please provide additional information regarding the permits you obtained for the work. Please ensure you have included the full name of the authority that approved the field site access and, if no permits were required, a brief statement explaining why.

3. Thank you for stating the following financial disclosure: “Deanship of Research and Graduate Studies at King Khalid University submitted the funding for this work through small group research under grant number RGP1/200/45.”

4. Thank you for stating the following in the Acknowledgments Section of your manuscript: “The authors extend their appreciation to the Deanship of Research and Graduate Studies at King Khalid University for funding this work through small group research under grant number RGP1/200/45.”

Please remove any funding-related text from the manuscript and let us know how you would like to update your Funding Statement. Currently, your Funding Statement reads as follows: “Deanship of Research and Graduate Studies at King Khalid University submitted the funding for this work through small group research under grant number RGP1/200/45.”

5. We note that your Data Availability Statement is currently as follows: “All relevant data are within the manuscript and its Supporting Information files.”

Reviewers' comments:

Reviewer's Responses to Questions

**Comments to the Author**

1. Is the manuscript technically sound, and do the data support the conclusions?

Reviewer #1: Yes

Reviewer #2: Partly

Reviewer #3: Partly

2. Has the statistical analysis been performed appropriately and rigorously? 

Reviewer #1: N/A

Reviewer #2: No

Reviewer #3: No

3. Have the authors made all data underlying the findings in their manuscript fully available?

Reviewer #1: Yes

Reviewer #2: Yes

Reviewer #3: Yes

4. Is the manuscript presented in an intelligible fashion and written in standard English?

Reviewer #1: Yes

Reviewer #2: No

Reviewer #3: No

5. Review Comments to the Author

Reviewer #1: The manuscript (Morphological characterization and (DNA) Barcoding identification of Ruellia spp. (Acanthaceae) in Saudi Arabia) is clear, unambiguous, and technically correct. The paper contains interesting results and is generally well-written and structured. The experiments were successful, and the data was well understood and modeled in detail.

In addition, the manuscript contains relevant paragraphs that have been discussed. The selection of the bibliography is appropriate to the content of the manuscript. Some minor errors appeared throughout the manuscript.

- The introduction is appropriate, but a few things need further improvements, especially the study hypothesis that should be added for the last five years.

- The phrase "recognized as the first taxonomic investigation of Ruellia species within the Kingdom of Saudi Arabia" is somewhat repetitive, as "the first taxonomic investigation" already implies this recognition.

- The enumeration of species and regions (e.g., "Abha, Aseer (24651)...") is cluttered. Consider rephrasing for clarity, perhaps using bullet points or breaking into more precise sentences.

- The abstract does not provide a good outline for the study.

- What is the limitation of this study?

- Arrange the keywords in alphabetical order.

- Cross-check the references in the text and reference cite.

- Ensure that the citation style remains consistent throughout the document. In the first section, "Migahid's (1974) study" differs from "(Migahid, 1974)" used later.

- Some sentences are pretty long and complex, which can be confusing. Breaking them into shorter sentences can improve comprehension. Additionally, ensuring consistent punctuation can aid in reader understanding.

- Conclusion: Improve this part concerning formulated objectives.

Reviewer #2: The manuscript ‘Morphological characterization and (DNA) Barcoding identification of Ruellia spp. (Acanthaceae) in Saudi Arabia’ [PONE-D-24-30210] focused on the genetic discrimination of Ruellia spp. Using DNA barcoding. I have some suggestions for improvement of the manuscript which is cited in the track changed MS. The English language, grammar, and punctuation need to be cross-checked by a native English speaker. References need to be thoroughly checked. Figures may be reduced to 4-5 with high-quality TIFF Images.

The MS may be thoroughly revised and resubmitted.

Reviewer #3: Peer Review Report for PLOS ONE

Manuscript title:

Morphological characterization and (DNA) Barcoding identification of Ruellia spp. (Acanthaceae) in Saudi Arabia

Summary statement:

Eight Ruellia species were collected from eight distinct natural habitats across different regions within KSA. four molecular barcoding markers were employed: Psbk-psbi, trnH-psbA, rbcL, and AtpFAtpH. The use of four DNA barcode markers facilitated the identification of five distinct species: R. tweediana, R. sp. SH2010, R. carolinensis, R. simplex, and R. patula.

Authors planned to:

Identify the various species within the genus Ruellia using both morphological characteristics and DNA barcoding methods.

It is concluded that:

The dominant Ruellia species in Saudi Arabia and demonstrated the reliability of DNA barcode markers for species identification.

These species are recorded for the first time in Saudi Arabia and represent the

first record.

Title:

The title of the paper is informative and relevant it is reflect the MS contents.

Abstract:

Sufficiently informative, covering all sections of the MS except methodology and clearly summarize the background, the aim of the study, results and conclusion. It reflects the article content. So, authors should supply a short statement describing the methods that was followed in the MS.

Introduction:

It describe what the author hoped to achieve accurately; the research question is outlined and justified and clearly state the problem being investigated. It summarizes relevant research to provide context, and explain what other authors findings.

Author/s gave what is already known about the use of DNA barcoding as an effective tool to identify diverse Ruellia species across various habitats

The goal of present study is to apply molecular approaches for the identification of genus Ruellia using both morphological characteristics and DNA barcoding methods.

Methodology:

The description of materials and methods insufficiently informative to allow replication of the experiment. The statistical methods not mentioned in the MS.

Results:

The experimental results are insufficient to justify the conclusions. Results poor presented.

Tables and figures

Tables not well presented

Figures lack scale bars

Discussion and conclusions:

The discussion and conclusion are not supported by the results.

References:

Relevant, adequate and properly chosen but need revision and to be up-to date because the recent one is at 2020.

Originality

The article is insufficiently novel but interesting to warrant publication.

Overall

The study is informative but lack novelty, I recommend it may be accepted for publishing after major revisions.

6. PLOS authors have the option to publish the peer review history of their article (what does this mean?). If published, this will include your full peer review and any attached files.

Reviewer #1: No

Reviewer #2: **Yes: **MRS

Reviewer #3: **Yes: **Usama K. Abdel-Hameed

---

## [Author Response · Author response to Decision Letter 0]

6 Nov 2024

Thanks for providing us the opportunity to revise the manuscript “Morphological characterization and DNA barcoding of Ruellia sp in Saudi Arabia” in Plos-one. We welcome the comprehensive suggestions and positive recommendations. All comments have been processed and responded to in a separate file labeled 'Response to Reviewers'.

---

## [Decision Letter · Decision Letter 1]

2 Dec 2024

Morphological characterization and DNA barcoding of Ruellia sp. in Saudi Arabia

PONE-D-24-30210R1

Dear Dr. ALSHEHRI,

We’re pleased to inform you that your manuscript has been judged scientifically suitable for publication and will be formally accepted for publication once it meets all outstanding technical requirements.

Kind regards,

Abul Khayer Mohammad Golam Sarwar

Academic Editor

PLOS ONE

Additional Editor Comments (optional):

Reviewers' comments:

Reviewer's Responses to Questions

**Comments to the Author**

1. If the authors have adequately addressed your comments raised in a previous round of review and you feel that this manuscript is now acceptable for publication, you may indicate that here to bypass the “Comments to the Author” section, enter your conflict of interest statement in the “Confidential to Editor” section, and submit your "Accept" recommendation.

Reviewer #1: All comments have been addressed

Reviewer #3: All comments have been addressed

2. Is the manuscript technically sound, and do the data support the conclusions?

Reviewer #1: Yes

Reviewer #3: Yes

3. Has the statistical analysis been performed appropriately and rigorously? 

Reviewer #1: Yes

Reviewer #3: Yes

4. Have the authors made all data underlying the findings in their manuscript fully available?

Reviewer #1: Yes

Reviewer #3: Yes

5. Is the manuscript presented in an intelligible fashion and written in standard English?

Reviewer #1: Yes

Reviewer #3: Yes

6. Review Comments to the Author

Reviewer #1: The authors have addressed my concerns. They have made a substantial modification to the manuscript. I recommend for publication.

best regards.

Reviewer #3: Thank you very much for your positive response as you mentioned in the comment reply, you revised the materials & methods section and add the statistical methods. In addition, attach the

Numerical analysis file, revised the result and conclusion section, revised the tables presentation.

We already have a scale bar for figures 3, 4 and 5, evised the discussion and aligned with results

and conclusion section, Updated with references published 2022و

7. PLOS authors have the option to publish the peer review history of their article (what does this mean?). If published, this will include your full peer review and any attached files.

Reviewer #1: No

Reviewer #3: **Yes: **Usama K. Abdel-Hameed

---

## [Editor Report · Acceptance letter]

13 Dec 2024

PONE-D-24-30210R1 

PLOS ONE

Dear Dr. ALSHEHRI, 

I'm pleased to inform you that your manuscript has been deemed suitable for publication in PLOS ONE. Congratulations! Your manuscript is now being handed over to our production team.

Kind regards, 

on behalf of

Professor Abul Khayer Mohammad Golam Sarwar 

Academic Editor

PLOS ONE